# Expert load matters: operating networks at high accuracy and low manual effort

**Sara Sangalli**
Computer Vision Lab
ETH Zürich
sara.sangalli@vision.ee.ethz.ch

**Ertunc Erdil**
Computer Vision Lab
ETH Zürich

**Ender Konukoglu**
Computer Vision Lab, ETH Zürich
The LOOP Zürich – Medical Research Center, Zürich, Switzerland

## Abstract

In human-AI collaboration systems for critical applications, in order to ensure minimal error, users should set an operating point based on model confidence to determine when the decision should be delegated to human experts. Samples for which model confidence is lower than the operating point would be manually analysed by experts to avoid mistakes. Such systems can become truly useful only if they consider two aspects: models should be confident only for samples for which they are accurate, and the number of samples delegated to experts should be minimized. The latter aspect is especially crucial for applications where available expert time is limited and expensive, such as healthcare. The trade-off between the model accuracy and the number of samples delegated to experts can be represented by a curve that is similar to an ROC curve, which we refer to as *confidence operating characteristic (COC)* curve. In this paper, we argue that deep neural networks should be trained by taking into account both accuracy and expert load and, to that end, propose a new complementary loss function for classification that *maximizes the area under this COC curve*. This promotes simultaneously the increase in network accuracy and the reduction in number of samples delegated to humans. We perform experiments on multiple computer vision and medical image datasets for classification. Our results demonstrate that the proposed loss improves classification accuracy and delegates less number of decisions to experts, achieves better out-of-distribution samples detection and on par calibration performance compared to existing loss functions.[1]

## 1 Introduction

Artificial intelligence (AI) systems based on deep neural networks have achieved state-of-the-art results by reaching or even surpassing human-level performance in many predictive tasks [6; 33; 2; 36]. Despite the great potential of neural networks for automation, there are pitfalls when using them in a fully automated setting, especially pertinent for safety-critical applications, such as healthcare [17; 32; 34]. Human-AI collaboration aims at remedying such issues by keeping humans in the loop and building systems that take advantage of both [29]. An example of such human-AI collaboration is hate speech detection for social media [3], where neural networks could reduce the load of manual analysis of contents required by humans. Healthcare is another relevant application [5; 22], . For

---

[1]Code is available at: https://github.com/salusanga/aucoc_loss.

37th Conference on Neural Information Processing Systems (NeurIPS 2023).

example, a neural network trained to predict whether a lesion is benign or malignant should leave the decision to human doctors if it is likely to make an error [15]. The doctors' domain knowledge and experience could be exploited to assess such, possibly, ambiguous cases and avoid mistakes.

A simple way of building collaboration between a network and a human expert is delegating the decisions to the expert when the network's confidence score for a prediction is lower than a threshold, which we refer to as "operating point". It is clear that the choice of the operating point can only be done through a trade-off between the network performance on the automatically analysed samples, i.e., the number of errors expected by the algorithm, and the number of delegated samples, i.e., experts' workload. The latter is crucial especially for applications where expert time is limited and expensive. For example, in medical imaging, the interpretation of more complex data requires clinical expertise and the number of available experts is limited, especially in low income countries [17]. Hence, predictive models that can analyse a large portion of the samples at high accuracy and identify the few samples that should be delegated to human experts would naturally be more useful with respect to this trade-off.

It is possible to evaluate the performance of a predictive model taking simultaneously into account the accuracy and the number of samples that requires manual assessment from a human expert with a performance curve reminiscent of Receiver Operating Characteristic (ROC) curves, as illustrated in Fig. 1a. We will refer to this performance curve as *Confidence Operating Characteristics (COC)* as it is similar to the ROC curve. A COC curve plots for a varying threshold on algorithm confidence, i.e., operating point, the accuracy of a model on the samples on which the algorithm is more confident than the threshold versus the number of samples remaining below the threshold. The former corresponds to the accuracy of the model on automatically analysed samples while the latter corresponds to the amount of data delegated to the human expert for analysis. In an ROC curve a balance is sought after between Sensitivity and Specificity of a predictive model, while a COC curve can be used by domain experts, such as doctors, to identify the most suitable balance between *the accuracy on the samples that are automatically analysed and the amount of data delegated to be re-examined by a human* for the specific task. Variations of this curve have been used to evaluate the performance of automated industrial systems [7].

In this paper, we focus on the trade-off between model accuracy and the amount of samples delegated to domain experts based on operating points on model confidence. Specifically, our goal is to obtain better trade-off conditions to improve the interaction between the AI and the human expert. To this end, we propose a new loss function for multi-class classification, that takes into account both of the aspects by maximizing the area under COC (AUCOC) curve. This enforces the simultaneous increase in neural network accuracy on the samples not analysed by the expert and the reduction in human workload. To the best of our knowledge, this is the first paper to include the optimization of such curve during the training of a neural network, formulating it in a differentiable way. We perform experiments on two computer vision and three medical image datasets for multi-class class classification. We compare the proposed complementary AUCOC loss with the conventional loss functions for training neural networks as well as network calibration methods. The results demonstrate that our loss function complements other losses and improved both accuracy and AUCOC. Additionally, we evaluate network calibration and out-of-distribution (OOD) samples detection performance of networks trained with different losses. The proposed approach was also able to consistently achieve better OOD samples detection and on par network calibration performance.

## 2   Related Work

For the performance analysis of human-AI collaborative systems, confidence operating characteristics (COC) curves can be employed, which plot network accuracy on accepted samples against manual workload of a human expert, e.g as in [7]. While such curves have been used, to the best of our knowledge, we present the first work that defines a differentiable loss based on COC curve, in order to optimize neural networks to take into account simultaneously accuracy and experts' load for a human-AI collaborative system. Thus, there is no direct literature with which we can compare.

The underlying assumption in deciding which sample to delegate to human expert based on a threshold on confidence scores is that these scores provided by deep learning models indicate how much the predictions are likely to be correct or incorrect. However, the final softmax layer of a network does not necessarily provide real probabilities of correct class assignments. In fact, modern deep neural

networks that achieve state-of-the-art results are known to be overconfident even in their wrong predictions. This leads to networks that are not well-calibrated, i.e., the confidence scores do not properly indicate the likelihood of the correctness of the predictions [8]. Network calibration methods mitigate this problem by calibrating the output confidence scores of the model, making the above mentioned assumption hold true. Thus, we believe that the literature on network calibration methods is the closest to our setting because they also aim at improving the interaction between human and AI, by enforcing correlation between network confidence and accuracy, so that confidence can be used to separate samples where networks' predictions are not reliable and should be delegated to human experts. Calibration methods aim to get models that are highly accurate in the samples they are confident, but not the problem of minimising the number of samples delegated to human experts, contrary to the loss proposed here.

Guo et al. [8] defines the calibration error as the difference in expectation between accuracy and confidence in each confidence bin. One category of calibration methods augments or replaces the conventional training losses with another loss to explicitly encourage reducing the calibration error. Kumar et al. [21] propose the MMCE loss by replacing the bins with kernels to obtain a continuous distribution and a differentiable measure of calibration. Karandikar et al. [16] propose two losses for calibration, called Soft-AvUC and Soft-ECE, by replacing the hard confidence thresholding in AvUC [18] and binning in ECE [8] with smooth functions, respectively. All these three functions are used as a secondary loss along with conventional losses such as cross-entropy. Mukhoti et al. [26] find that Focal Loss (FL) [23] provides inherently more calibrated models, even if it was not originally designed for this, as it adds implicit weight regularisation. The authors further propose Adaptive Focal Loss (AdaFL) with a sample-dependent schedule for the choice of the hyperparameter $\gamma$. The second category of methods are post-hoc calibration approaches, which rescale model predictions after training. Platt scaling [31] and histogram binning [40] fall into this class. Temperature scaling (TS) [8] is the most popular approach of this group. TS scales the logits of a neural network, dividing them by a positive scalar, such that they do not saturate after the subsequent softmax activation. TS can be used as a complementary method and it does not affect model accuracy, while significantly improving calibration. A recent work by Gupta et al. [9] fits a spline function to the empirical cumulative distribution to re-calibrate post-hoc the network outputs. They also present a binning-free calibration measure inspired by the Kolmogorov-Smirnov (KS) statistical test. Lin et al. [24] propose a Kernel-based method on the penultimate-layer latent embedding using a calibration set.

## 3 Methods

In this section, we illustrate in detail the *Confidence Operating Characteristics (COC)* curve. Then, we describe the proposed complementary cost function to train neural networks for classification: *the area under COC (AUCOC) loss (AUCOCLoss)*.

### 3.1 Notation

Let $D = \langle (x_n, y_n) \rangle_{n=1}^N$ denote a dataset composed of $N$ samples from a joint distribution $\mathcal{D}(\mathcal{X}, \mathcal{Y})$, where $x_n \in \mathcal{X}$ and $y_n \in \mathcal{Y} = \{1, 2, ..., K\}$ are the input data and the corresponding class label, respectively. Let $f_\theta(y|x)$ be the probability distribution predicted by a classification neural network $f$ parameterized by $\theta$ for an input $x$. For each data point $x_n$, $\hat{y}_n = \operatorname{argmax}_{y \in \mathcal{Y}} f_\theta(y|x_n)$ denotes the predicted class label, associated to a *correctness score* $c_n = \mathbb{1}(\hat{y}_n = y_n)$ and to a *confidence score* $r_n = \max_{y \in \mathcal{Y}} f_\theta(y|x_n)$, where $r_n \in [0, 1]$ and $\mathbb{1}(.)$ is an indicator function. $\mathbf{r} = [r_1, ... r_N]$ represents the vector containing all the predicted confidences for a set of data points, e.g., a batch. $p(r)$ denotes the probability distribution over $r$ values (confidence space). We assume a human-AI collaboration system where samples with confidence $r$ lower than a *threshold* $r_0$ would be delegated to a human expert for assessment.

### 3.2 Confidence Operating Characteristics (COC) curve

Our first goal is to introduce an appropriate evaluation method to assess the trade-off between a neural network's prediction accuracy and the number of samples that requires manual analysis from a domain expert. We focus on the COC curve, as it provides practitioners with flexibility in the choice of the operating point, similarly to the ROC curve.

### 3.2.1 x-y axes of the COC curve

To construct the COC curve, first, we define a *sliding threshold* $r_0$ over the space of predicted confidences $r$. Then, for each threshold $r_0$, we calculate the portion of samples that are delegated to human expert and the accuracy of the network on the remaining samples for the threshold $r_0$, which form the **x-axis** and **y-axis** of a COC curve, respectively. These axes are formulated as follows

$$x - \text{axis}: \ \tau_0 = p(r < r_0) = \int_0^{r_0} p(r)dr, \qquad y - \text{axis}: \mathbb{E}[c|r \geq r_0] \qquad (1)$$

For each threshold level $r_0$, $\tau_0$ represents the portion of samples whose confidence is lower than that threshold, i.e., the amount of the samples that are delegated to the expert. $\mathbb{E}[c|r \geq r_0]$ corresponds to the expected value of the correctness score $c$ for all the samples for which the network's confidence is equal or larger than $r_0$, i.e., accuracy among the samples for which network prediction will be used. This expected value, i.e., y-axis, can be computed as

$$\mathbb{E}[c|r \geq r_0] = \frac{1}{1 - \tau_0} \int_{r_0}^1 \mathbb{E}[c|r]p(r)dr. \qquad (2)$$

We provide the derivation of Eq. 2 in the Appendix A.

### 3.2.2 Area under COC curve

Like the area under ROC curve, area under COC curve (AUCOC) is a global indicator of the performance of a system. Higher AUCOC indicates lower number of samples delegated to human experts or/and higher accuracy for the samples that are not delegated to human experts but analysed only by the network. Lower AUCOC on the other hand, indicates higher number of delegations to human experts or/and lower accuracy on the samples analysed only by the network. Further explaination is reported in Appendix I. It can be computed by integrating the COC curve over the whole range of $\tau_0 \in [0, 1]$:

$$AUCOC = \int_0^1 \mathbb{E}[c|r \geq r_0]d\tau_0 = \int_0^1 \left\{ \int_{r_0}^1 \mathbb{E}[c|r]p(r)dr \right\} \frac{d\tau_0}{1 - \tau_0} \qquad (3)$$

Further details are provided in Section 3.5

### 3.3 AUCOCLoss: Maximizing AUCOC for training neural networks

As mentioned, in human-AI collaboration systems, higher AUCOC means a better model. Therefore, in this section we introduce a new loss function called AUCOCLoss that maximizes AUCOC for training classification neural networks.

AUCOC's explicit maximization would enforce the reduction of the number of samples delegated to human expert while maintaining the accuracy level on the samples assessed only by the algorithm (i.e., keeping $\mathbb{E}[c|r \geq r_0]$ constant) and/or the improvement in the prediction accuracy of the samples analysed only by the algorithm while maintaining a particular amount of data to be delegated to the human (i.e., keeping $\tau_0$ constant), as illustrated in Figure 1a.

We define our loss function to maximize AUCOC as

$$AUCOCLoss = -\log(AUCOC). \qquad (4)$$

We use the negative logarithm as AUCOC lies in the interval $[0, 1]$, which corresponds to AUCOCLoss $\in [0, \inf]$ which is suitable for minimizing cost functions. The dependence of AUCOC on the network parameters may not be obvious from the formulation given in Eq. 3. Indeed, this dependence is hidden in how $p(r)$ and $\mathbb{E}[c|r]$ are estimated as we show next.

### 3.3.1 Kernel density estimation for AUCOC

We need to formulate AUCOC in a differentiable way in order for it to be incorporated in a cost function for the training of a neural network. For this purpose, we use kernel density estimation (KDE) on confidence predictions $r_n$ for training samples to estimate $p(r)$ used in Eq. 3

$$p(r) \approx \frac{1}{N} \sum_{n=1}^N K(r - r_n) \qquad (5)$$

where $K$ is a Gaussian kernel and we choose its bandwidth using Scott's rule of thumb [35]. Then, the other terms in Eq. 3, namely $\mathbb{E}[c|r]p(r)$ and $\tau_0$, are estimated as

$$\mathbb{E}[c|r]p(r) \approx \frac{1}{N}\sum_{n=1}^{N} c_n K(r - r_n), \qquad \tau_0 \approx \frac{1}{N}\int_0^{r_0}\sum_{n=1}^{N} K(r - r_n)dr. \qquad (6)$$

Note that $r_n = \max_{y \in \mathcal{Y}} f_\theta(y|x_n)$, from where the dependency of AUCOC on $\theta$ stems. Further, note that $\tau_0$ is the x-axis of the COC curve. The AUCOC is defined by integrating over the $\tau_0$ values, and therefore $\tau_0$ should not depend on the network parameters, i.e., its derivative with respect to $\theta$ should be equal to zero. We can write the derivative using Leibniz integral rule as follows:

$$\frac{d\tau_0}{d\theta} = \int_0^{r_0} \frac{dp(r)}{d\theta} dr + p(r_0)\frac{dr_0}{d\theta} = 0 \qquad (7)$$

Then, the constraint that $\tau_0$ should not depend on $\theta$ can be enforced explicitly by deriving the derivative $dr_0/d\theta$ from Eq. 7 as follows:

$$\frac{dr_0}{d\theta} = -\frac{\int_0^{r_0} \frac{dp(r)}{d\theta} dr}{p(r_0)} \qquad (8)$$

where $p(r)$ is implemented as in Eq. 5. Derivations are provided in the Appendix.

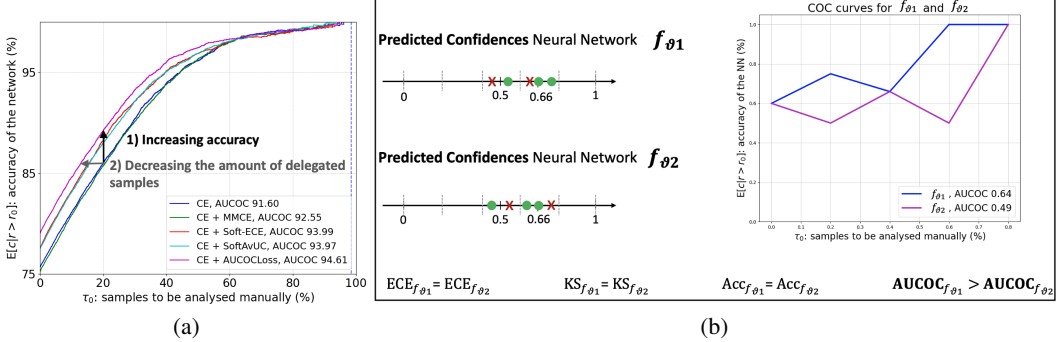

Figure 1: **(a)** shows how to improve AUCOC, 1) increasing the accuracy of the network and/or 2) decreasing the amount of data to be analysed by the domain expert. The pink curve has higher AUCOC than the blue one. **(b)** illustrates a toy example where two models have the same accuracy, ECE with 5 bins and KS. However, they have different AUCOC values due to different ordering of correctly and incorrectly classified samples according to the assigned confidence by the network.

### 3.4 Toy example: added value by AUCOC

In this section, we demonstrate the added value of assessing the performance of a predictive model using COC curve and AUCOC through a toy example. We particularly compare with the widely used expected calibration error (ECE) [8], a binning-free calibration metric called Kolmogorov-Smirnov (KS) [9] and classification accuracy.

Assume we have two classification models $f_{\theta_1}$ and $f_{\theta_2}$ and they yield confidence scores and predictions for 5 samples as shown in Figure 1b. The green circles denote the predicted confidences for correctly classified samples, while the red crosses the confidences of the misclassified ones.

**ECE:** ECE divides the confidence space into bins, computes the difference between the average accuracy and confidence for each bin, and returns the average of the differences as final measure of calibration error. If we divide the confidence space into 5-bins, as indicated with the gray dotted lines in the confidence spaces of $f_{\theta_1}$ and $f_{\theta_2}$, ECEs computed for the both models will be identical. Furthermore, a similar situation can be constructed for any number of bins.
**KS:** KS is a binning-free metric, so it is less prone to binning errors than ECE. However, one can prove that there exist some confidence configurations for which the two models also report the same

KS, in spite of it being a binning-free metric. This happens, for example, if the confidence values are (from left to right): 0.45, 0.55, 0.65, 0.70, 0.75.
**Accuracy:** These models have equal classification accuracy, 3/5 for both.

Therefore, looking at these three performance metrics, it is not possible to choose one model over the other since $f_{\theta_1}$ and $f_{\theta_2}$ perform identically.

**AUCOC:** On the contrary, the AUCOC is larger for $f_{\theta_1}$ than for $f_{\theta_2}$, as shown in Figure 1b. The difference in AUCOC is due to the *different ranking* of correctly and incorrectly classified samples with respect to confidence values. It does not depend on a particular binning nor exact confidence values, but only on the ranking. By looking at the AUCOC results, one would prefer $f_{\theta_1}$ compared to $f_{\theta_2}$. Indeed, $f_{\theta_1}$ is a better model than $f_{\theta_2}$ because it achieves either equal or better accuracy than $f_{\theta_2}$, for the same amount of data to be manually examined. Changing point of view, $f_{\theta_1}$ delegates either equal or lower number of samples to experts for the same accuracy level.
**Comparison of AUCOC and calibration:** As also demonstrated with the toy example, AUCOC and calibration assess different aspects of a model. The former is a rank-based metric with respect to the ordering of the predictive confidences of correctly and incorrectly classified samples. The latter instead assess the consistency between the accuracy of the network and the predictive confidences, i.e. it is sensitive to the numerical values of the confidences themselves.

### 3.5 Implementation Details

**Construction of the COC curve, thresholds $r_0$ and operating points:** To provide flexibility in the selection of COC operating points, we need to cover the entire range $[0, 1]$ of $\tau_0$ values. As a consequence, the thresholds $r_0$ need to span the confidence range of the predictions $r_n$ of the neural network. A natural choice for such thresholds is employing the predictions $r_n$ themselves. This spares us from exploring the confidence space with arbitrarily fine-grained levels. First, we sort the confidences $\mathbf{r}$ of the whole dataset (or batch) in ascending order. Each predicted confidence is then selected as threshold level $r_0$ for the vector $\mathbf{r}$, corresponding to a certain $\tau_0$ (x-value). Subsequently, $\mathbb{E}[c|r \geq r_0]$ (y-value) is computed. Note that setting the threshold to each $r_n$ in the sorted array corresponds to going through $\tau_0 = [1/N, 2/N, \ldots, (N-1)/N, 1]$ for $N$ samples one by one in order.

**Modelling correctness:** Instead of using $\mathbb{E}[c|r]p(r) \approx \frac{1}{N}\sum_{n=1}^{N} c_n K(\|r - r_n\|)$ as given in Eq. 6, we approximate it as $\mathbb{E}[c|r]p(r) \approx \frac{1}{N}\sum_{n=1}^{N} r^* K(\|r - r_n\|)$ where $r_n^* = f_\theta(y_n|x_n)$ is the confidence of the correct class for a sample $n$. The main reason is that the gradient of the misclassified samples becomes zero because $c_n$ is zero when a sample $x_n$ is not classified correctly. To deal with this issue we replace the correctness score $c_n$, which can be either 0 or 1, with $r_n^*$ which can take continuous values between 0 and 1, following Yin et al. [39]. With this new approximation, we can back-propagate through misclassified samples and we found that this leads to better results.

**Use as secondary loss:** We observed in our experiments that using AUCOCLoss alone to train a network leads to very slow convergence with the existing optimization techniques. We believe this is due to the fact that the AUCOCLoss is a $-log(\sum_n z_n)$ with $z_n \in [0, 1]$. Optimization of such a form with gradient descent is slow because the contribution of increasing low $z_n$'s to the loss function is small. In contrast, cross-entropy is a $-\sum_n \log z_n$, where contribution of increasing low $z_n$'s to the loss is much larger, hence gradient-based optimization is faster. One can in theory create an upper bound to AUCOC loss by pulling the $\log$ inside the sum, as we show in the Appendix. However, when $z_n \in [0, 1]$ this upper bound is very loose and minimizing the upper bound not necessarily correspond to minimizing the AUCOC loss, hence does not maximize AUCOC. On the contrary, when AUCOCLoss is complements a primary cost that is faster to optimize, such as cross-entropy, it is improves over the primary loss and lead to the desired improved AUCOC while preserving the accuracy. This is obtained within the same amount of epochs and without ad-hoc fine-tuning the training hyper-parameters for the secondary loss.

## 4 Experiments

In this section, we present our experimental evaluations on multi-class image classification tasks. We performed experiments on five datasets. We experimented with CIFAR100 [19] and Tiny-ImageNet, a subset of ImageNet [4], following the literature on network calibration. Further, we used two publicly

available medical imaging datasets, DermaMNIST and RetinaMNIST [38], since medical imaging is an application area where expert time is limited and expensive. Due to space reasons we report results on a third medical dataset, TissueMNIST [38], in Appendix D, as well as information about the datasets in Appendix C.

**Loss functions:** We compared AUCOCLoss (referred to as AUCOCL in the tables) with different loss functions, most of which are designed to improve calibration performance while preserving accuracy: cross-entropy (CE), focal-loss (FL) [23], adaptive focal-loss (AdaFL) [26], maximum mean calibration error loss (MMCE) [21], soft binning calibration objective (S-ECE) and soft accuracy versus uncertainty calibration (S-AvUC) [16]. We optimized MMCE, S-ECE, and S-AvUC losses jointly with a primary loss for which we used either CE or FL, consistently with the literature [21; 16]. The same is done for AUCOCLoss, applying KDE batch-wise during the training.

**Evaluation metrics:** To evaluate the performance of the methods, we used classification accuracy and AUCOC. Classification accuracy is simply the ratio between the number of correct samples over the total number of samples, and AUCOC is computed using Eq. 3. We also report some examples of operating points of COC curve - given a certain accuracy, we show the corresponding expert load ($\tau_0$ @acc), i.e., percentage of samples that need to be analyzed manually, on the COC curves.

Since we compared AUCOCLoss mostly with losses for network calibration, we also assessed the calibration performance of the networks, even though calibration is not a direct goal of this work. To this end, we used the following metrics: the widely employed equal-mass expected calibration error (ECE) [28] with 15 bins, the binning-free Brier score [1] and Kolmogorov-Smirnov (KS) score [9] and the class-wise ECE (cwECE) [20]. The evaluation is carried out post temperature scaling (TS) [8], as it has been proved to be always beneficial for calibration.

**Hyperparameters:** In addition to the common hyperparameters for all losses, selected as specified in Appendix C, there are also specific ones that need to be tuned for some of them. In this case, we used the best hyperparameter settings reported in the original papers. In cases where the original paper did not report the specific values, we carried out cross-validation and selected the setup that provided the best performance on the validation set. We also selected the weighting factor for AUCOCLoss in the same way. We found that optimal weighting values for AUCOCLoss all fell between 1 and 10. We found empirically that models check-pointed using ECE provided very poor results. Networks check-pointed using either accuracy or AUCOC provided comparable outcome with respect to accuracy, therefore we reported results on AUCOC as they provided the best overall performance.

**OOD experiments:** We evaluate the out-of-distribution (OOD) samples detection performance of all methods since OOD detection is crucial for a reliable AI system and it is a common experiment in the network calibration literature. The commonly used experimental setting in this literature is using CIFAR100-C [12] (we report results for CIFAR100 with Gaussian noise in the main paper and the average over all the perturbations in the Appendix) and SVHN [27] as OOD datasets, while the network is trained on CIFAR100 (in-distribution). We evaluated the OOD detection performance of all methods using Area Under the Receiver Operating Characteristics (AUROC) curve, with MSP [13], ODIN [14], MaxLogit [11] and EBM [25].

**Class imbalance experiments:** Finally, in Appendix E we report results for accuracy and AUCOC on imbalanced datasets, being class imbalance present in many domains, such as medical imaging. We report results on the widely used Long-Tailed CIFAR100 (CIFAR100-LT) [37; 42] with controllable degrees of data imbalance ratio to control the distribution of training set. We trained with three levels of imbalance ratio, namely 100, 50, 10.

Further details on hyper-parameter settings are provided in the the Appendix.

## 5 Results

First, we report the results for accuracy and COC-related metrics. Then, we report results for calibration, even though this is not an explicit goal of this work. Bold results indicate the methods that performed best for each metric, underlined results are the second best. ↑ means the higher the better for a metric, while ↓ the lower the better. The experiments results are averaged over three runs.

Table 1: Test results on the natural datasets CIFAR100 and Tiny-Imagenet. We report AUCOC, accuracy for both, $\tau_0$ at 90% and 95% accuracy for CIFAR100 and $\tau_0$ at 65% and 75% accuracy for Tiny-ImageNet, as the initial accuracy is also lower. In bold the best result for each metric, underlined the second best. AUCOCL improves, to varying degrees, the baselines in all metrics.

| Dataset | CIFAR100 | | | | Tiny-ImageNet | | | |
|---|---|---|---|---|---|---|---|---|
| | | | $\tau_0 \downarrow$ @ acc. | | | | $\tau_0 \downarrow$ @ acc. | |
| Loss funct. | AUCOC ↑ | Acc. ↑ | 90% | 95% | AUCOC ↑ | Acc. ↑ | 65% | 75% |
| CE | 91,43 | 75,71 | 29,03 | 44,61 | 72,56 | 47,39 | 39,88 | 56,29 |
| FL ($\gamma$=3) | 93,91 | 77,83 | 24,71 | 40,48 | 73,12 | 47,71 | 38,40 | 55,08 |
| AdaFL53 | 93,89 | 77,64 | 25,08 | 40,74 | 73,19 | 47,81 | 38,56 | 55,20 |
| CE+MMCE | 92,42 | 75,35 | 30,01 | 44,71 | 72,47 | 47,11 | 39,94 | 56,70 |
| FL+MMCE | 93,90 | 77,78 | 25,62 | 40,90 | 73,51 | 48,03 | 37,83 | 55,15 |
| CE+S-AvUC | 93,99 | 77,65 | 24,62 | 45,00 | 72,92 | 47,89 | 38,28 | 54,91 |
| FL+ S-AvUC | 93,97 | 77,64 | 25,97 | 40,82 | 74,30 | 48,69 | 35,83 | 53,21 |
| CE+S-ECE | 93,88 | 77,57 | 24,76 | 40,14 | 72,94 | 47,65 | 38,81 | 55,84 |
| FL+S-ECE | 93,41 | 76,69 | 28,13 | 43,29 | 72,61 | 47,40 | 39,94 | 56,71 |
| **CE+AUCOCL** | **94,49** | **78,94** | **21,60** | 36,73 | **74,56** | 49,10 | **34,78** | 52,01 |
| **FL+AUCOCL** | 94,18 | 78,31 | 23,50 | **36,68** | 74,30 | **49,19** | 34,85 | 53,15 |

Table 2: Test results on the medical datasets DermaMNIST and RetinaMNIST. We report AUCOC, accuracy for both, $\tau_0$ at 90% and 95% accuracy for DermaMNIST and $\tau_0$ at 65% and 75% accuracy for RetinaMNIST, as the initial accuracy is also lower. In bold the best result for each metric, underlined the second best. AUCOCL improves, to varying degrees, the baselines in all metrics.

| Dataset | DermaMNIST | | | | RetinaMNIST | | | |
|---|---|---|---|---|---|---|---|---|
| | | | $\tau_0 \downarrow$ @ acc. | | | | $\tau_0 \downarrow$ @ acc. | |
| Loss funct. | AUCOC ↑ | Acc. ↑ | 90% | 95% | AUCOC ↑ | Acc. ↑ | 65% | 75% |
| CE | 89,84 | 71,59 | 43,51 | 56,21 | 71,45 | 52,10 | 39,53 | 68,58 |
| FL ($\gamma$=3) | 90,50 | 72,64 | 40,63 | 53,90 | 68,57 | 52,25 | 44,25 | 58,25 |
| AdaFL53 | 90,11 | 73,10 | 40,78 | 55,86 | 68,85 | 48,58 | 48,67 | 62,00 |
| CE+MMCE | 89,71 | 70,99 | 45,82 | 58,07 | 69,18 | 48,50 | 42,92 | 69,13 |
| FL+MMCE | 89,34 | 71,72 | 46,81 | 59,10 | 67,08 | 50,33 | 47,50 | 78,91 |
| CE+S-AvUC | 89,67 | 71,51 | 43,04 | 57,09 | 68,15 | 51,42 | 42,71 | 62,29 |
| FL+ S-AvUC | 89,42 | 71,04 | 45,47 | 58,69 | 66,80 | 52,00 | 45,58 | 70,54 |
| CE+S-ECE | 89,54 | 71,46 | 43,48 | 57,82 | 71,40 | 52,05 | 39,45 | 55,83 |
| FL+S-ECE | 90,22 | 72,62 | 40,95 | 57,46 | 70,49 | 51,33 | 42,42 | 79,91 |
| **CE+AUCOCL** | 90,87 | 74,30 | 39,01 | **52,70** | **72,47** | 53,10 | **38,33** | 53,81 |
| **FL+AUCOCL** | **91,35** | **74,80** | **37,30** | 53,90 | 72,31 | **53,58** | 39,42 | 56,12 |

In Tables 1, 2 we present our results based on accuracy and AUCOC. The results for TissueMNIST can be found in the Appendix. We observed that complementing CE and FL with our loss, i.e., CE+AUCOCL and FL+AUCOCL, they were consistently better than the other losses in all experiments. To evaluate the significance of the accuracy and AUCOC results, we performed permutation test [30] between the best AUCOCLoss and the best baseline with 1000 rounds. In all the cases it provided a p score « 1%, therefore the differences in the models are steadily significant. The advantage of our model was even more apparent in amount of expert loads corresponding to specific accuracy levels. In particular, we measured the percentage of samples delegated to expert ($\tau_0$) at 90% and 95% accuracy for CIFAR100, DermaMNIST and TissueMNIST, and at 65% and 75% accuracy for Tiny-Imagenet and RetinaMNIST (as the initial accuracy is also much lower on these datasets). In all the experiments, to varying degrees, AUCOCLoss provided lower delegated samples than the baselines. Noticeably, while some of the baselines may be close to the results of AUCOCLoss for certain metrics, none of them were consistently close to AUCOCLoss across all the datasets. For example, when looking at the last two columns for CIFAR100 and Tiny-ImageNet respectively of Table 1, CE+MMCE is worse by 4% than AUCOCLoss on Tiny-ImageNet, but by around 9% in

Table 3: Test results on DermaMNIST and RetinaMNIST for calibration: expected calibration error (ECE), KS score, Brier score and class-wise ECE (cwECE), post TS. In bold and underlined respectively are the best and second best results for each metric. Noticeably, AUCOCL performs comparably to the baselines, even though it does not aim at improving calibration explicitly.

| Dataset | DermaMNIST | | | | RetinaMNIST | | | |
|---|---|---|---|---|---|---|---|---|
| Loss funct. | ECE↓ | KS↓ | Brier↓ | cwECE↓ | ECE↓ | KS↓ | Brier↓ | cwECE↓ |
| CE | **3,07** | 2,15 | 38,11 | 2,22 | 8,42 | 6,15 | 59,75 | 6,01 |
| FL ($\gamma$=3) | 4,24 | 1,77 | 36,55 | 2,03 | 13,61 | 10,87 | 63,32 | 8,19 |
| AdaFL53 | 3,88 | 1,95 | 36,19 | **1,66** | 13,63 | 11,12 | 65,64 | 8,47 |
| CE+MMCE | 3,59 | 2,73 | 38,96 | 2,20 | 11,58 | 9,75 | 62,18 | 6,87 |
| FL+MMCE | 3,65 | 2,29 | 38,91 | 2,34 | 12,49 | 10,54 | 65,72 | 7,81 |
| CE+S-AvUC | 3,79 | 2,69 | 38,28 | 2,21 | 8,37 | 7,01 | 64,36 | 6,91 |
| FL+ S-AvUC | 3,48 | 2,13 | 37,98 | 1,82 | 11,60 | 5,86 | 64,06 | 7,12 |
| CE+S-ECE | 3,23 | 2,73 | 38,39 | 1,92 | 9,44 | 5,88 | 59,84 | 5,37 |
| FL+S-ECE | 4,5 | 2,67 | 36,91 | 2,08 | 12,52 | 10,05 | 61,16 | 6,08 |
| **CE+AUCOCL** | 5,70 | **1,56** | 36,12 | 1,78 | **8,15** | **4,47** | **58,69** | **4,46** |
| **FL+AUCOCL** | 5,10 | 2,04 | **35,46** | 2,04 | 10,77 | 8,84 | 60,52 | 5,32 |

Table 4: Test AUROC(%) on OOD detection, training on CIFAR100 and testing on CIFAR100-C (Gaussian noise) and SVHN, using MSP, ODIN MaxLogit and EBM. Best and second best results are in bold and underlined.

| Dataset | **C100-C** AUROC↑ | | | | **SVHN** AUROC↑ | | | |
|---|---|---|---|---|---|---|---|---|
| Loss funct. | MSP | ODIN | MaxLogit | EBM | MSP | ODIN | MaxLogit | EBM |
| CE | 74,37 | 75,22 | 74,51 | 66,18 | 77,42 | 79,42 | 67,71 | 67,71 |
| FL ($\gamma$=3) | 75,12 | 75,35 | 73,43 | 72,83 | 76,61 | 77,08 | 66,69 | 66,42 |
| AdaFL53 | 74,53 | 74,68 | 71,29 | 70,58 | 80,3 | 81,28 | 66,94 | 66,73 |
| CE+MMCE | 74,67 | 74,53 | 70,67 | 70,17 | 75,79 | 77,39 | 66,34 | 66,23 |
| FL+MMCE | 74,42 | 74,39 | 70,01 | 68,70 | 77,57 | 77,41 | 66,97 | 66,60 |
| CE+S-AvUC | 73,63 | 73,62 | 72,16 | 72,05 | 78,04 | 78,90 | 67,93 | 67,96 |
| FL+ S-AvUC | 72,78 | 76,72 | 69,53 | 68,51 | 79,68 | 79,68 | 67,38 | 67,24 |
| CE+S-ECE | 73,29 | 73,18 | 71,59 | 71,36 | 77,02 | 77,69 | 67,98 | 68,01 |
| FL+S-ECE | 74,29 | 73,97 | 71,51 | 70,26 | 79,68 | 81,17 | 67,38 | 67,78 |
| **CE+AUCOCL** | 76,03 | **76,90** | **78,02** | **78,30** | **82,03** | **83,50** | 69,51 | **69,69** |
| **FL+AUCOCL** | **76,51** | 76,82 | 75,14 | 74,68 | 80,51 | 79,35 | **69,52** | 69,46 |

CIFAR100. Figure 1a shows examples of COC curves. Overall, the plot of AUCOCLoss lies above all the baselines, which is a desirable behavior as it corresponds to better operating points.

Even though the proposed loss was not designed to improve calibration, it provided on par performance compared to the other cost functions particularly designed for network calibration, as reported in Table 3 for DermaMNIST and RetinaMNIST and in the Appendix for the other datasets.

In OOD experiments reported in Table 4, we used the model trained on CIFAR100 and evaluated the OOD detection performance on CIFAR100-C (with Gaussian noise) and SVHN dataset. Results for CIFAR100-C averaged over all the perturbations are reported in the Appendix. We employed state-of-the-art OOD detectors, namely MSP, ODIN, MaxLogit and EBM to determine whether a sample is OOD or in-distribution. The bold results highlight the best results in terms of AUROC. On both OOD datasets, AUCOCLoss always provided the highest AUROC and in almost all the cases also the second best.

Class imbalance experiments on CIFAR100-LT are reported in the Appendix. AUCOCLoss obtains best results for both accuracy and AUCOC, with higher benefits at increasing imbalance.

Crucially, **CE+AUCOCL**, where the proposed loss is used jointly with CE, outperformed every baseline in accuracy, AUCOC and $\tau_0$ @acc. in all the experiments. It further outperformed all the baselines in OOD detection and class imbalance experiments, presented in the Appendix. Even

though the metric is not geared towards calibration, **CE+AUCOCL** yielded either the best or the second best calibration performance in the majority of the cases compared to the baselines.

In the Appendix, we explore how the performance of a model trained with AUCOCLoss varies, when changing batch size as it can be crucial for KDE-based methods. Even substantially lower batsch sizes do not have a considerable effect on the performance of the proposed method.

## 6  Conclusion

In this paper we proposed a new cost function for multi-class classification that takes into account the trade-off between a neural network's accuracy and the amount of data that requires manual analysis from a domain expert, by maximizing the area under COC (AUCOC) curve. Experiments on multiple computer vision and medical image datasets suggest that our approach improves the other methods in terms of both accuracy and AUCOC, where the latter was expected by design, provides comparable calibration metrics, even though the loss does not aim to improve calibration explicitly and outperforms the baselines in OOD detection.

While we presented COC and AUCOCLoss for multi-class classification, extensions to other tasks are possible future work as well as investigating different performance metrics to embed in the $y$-axis of COC. Moreover, aware of potential problems with KDE at the boundaries, i.e., boundary bias, we explored corrections like reflection method, which did not provide major improvements, but we will further investigate. We believe that this new direction of considering expert load in human-AI system is important and AUCOCLoss will serve as a baseline for future work.

**Limitations:** As described in Section 3.5, AUCOCLoss alone empirically leads to slow convergence, due to the small contribution of $-log(\sum_n z_n)$ with $z_n \in [0, 1]$ in its formulation. Therefore, we recommend to use it as a secondary loss, to successfully complement existing cost functions.

## 7  Acknowledgments

This study was financially supported by: 1. The LOOP Zürich – Medical Research Center, Zurich, Switzerland, 2. Personalized Health and Related Technologies (PHRT), project number 222, ETH domain and 3. Clinical Research Priority Program (CRPP) Grant on Artificial Intelligence in Oncological Imaging Network, University of Zürich.

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

# A Derivations

In this Section we provide the derivation of AUCOC and its gradient formulation.

## A.1 Derivation of AUCOC

AUCOC is defined as:

$$AUCOC = \int_0^1 \mathbb{E}[c|r \geq r_0]d\tau_0 = \int_0^1 \left\{ \int_{r_0}^1 \mathbb{E}[c|r]p(r)dr \right\} \frac{d\tau_0}{1-\tau_0}$$

The y-axis in COC curve is expressed mathematically by:

$$\mathbb{E}[c|r \geq r_0] = \sum p(c|r \geq r_0)c = \sum \frac{p(c, r \geq r_0)}{p(r \geq r_0)}c = \sum \frac{\int_{r_0}^1 p(c,r)dr}{\int_{r_0}^1 p(r)dr}c$$

$$= \sum \frac{\int_{r_0}^1 p(c|r)p(r)dr}{\int_{r_0}^1 p(r)dr}c = \frac{\int_{r_0}^1 \sum p(c|r)cp(r)dr}{\int_{r_0}^1 p(r)dr} = \frac{\int_{r_0}^1 \mathbb{E}[c|r]p(r)dr}{\int_{r_0}^1 p(r)dr} =$$

$$= \frac{\int_{r_0}^1 \mathbb{E}[c|r]p(r)dr}{1-\tau_0}$$

The x-axis in COC curve is expressed mathematically by:

$$\tau_0 = p(r < r_0) = \int_0^{r_0} p(r)dr$$

Using the $\tau_0$ formulation, we can rewrite the y-axis as

$$\mathbb{E}[c|r \geq r_0] = \frac{\int_{r_0}^1 \mathbb{E}[c|r]p(r)dr}{1-\tau_0}$$

Maximizing the area under this curve, over $\tau_0 \in [0, 1]$ corresponds to

$$\max \mathcal{A} = \max \int_0^1 \left\{ \int_{r_0}^1 \mathbb{E}[c|r]p(r)dr \right\} \frac{d\tau_0}{1-\tau_0}$$

Let us assume to use a Gaussian kernel with this expression:

$$K(||r - r_n||) = \frac{1}{\sqrt{2\pi\alpha}} \cdot \exp\left(-\frac{(r-r_n)^2}{2\alpha^2}\right) \tag{9}$$

Developing the equation, the area calculation becomes:

$$\mathcal{A} = \int_0^1 \left\{ \int_{r_0}^1 \mathbb{E}[c|r]p(r)dr \right\} \frac{d\tau_0}{1-\tau_0} =$$

$$= \int_0^1 \left\{ \int_{r_0}^1 \frac{1}{N}\sum_{n=1}^N \mathbf{1}(c_n)K(||r-r_n||)dr \right\} \frac{d\tau_0}{1-\tau_0} =$$

$$= \int_0^1 \left\{ \int_{r_0}^1 \frac{1}{N}\sum_{n=1}^N \mathbf{1}(c_n)\frac{1}{\sqrt{2\pi\alpha}} \cdot \exp\left(-\frac{(r-r_n)^2}{2\alpha^2}\right)dr \right\} \frac{d\tau_0}{1-\tau_0} =$$

$$= \int_0^1 \left\{ \frac{1}{N}\sum_{n=1}^N \mathbf{1}(c_n)\int_{r_0}^1 \frac{1}{\sqrt{2\pi\alpha}} \cdot \exp\left(-\frac{(r-r_n)^2}{2\alpha^2}\right)dr \right\} \frac{d\tau_0}{1-\tau_0} =$$

$$= \int_0^1 \left\{ \frac{1}{N}\sum_{n=1}^N \mathbf{1}(c_n)\left(ndtr\left(\frac{1-r_n}{\sqrt{cov}}\right) - ndtr\left(\frac{r_0-r_n}{\sqrt{cov}}\right)\right) \right\} \frac{d\tau_0}{1-\tau_0} =$$

$$= \sum_{k=1}^{\#thresh} \frac{f(\tau_{0,k-1}, r_{0,k-1}) + f(\tau_{0,k}, r_{0,k})}{2}(\tau_{0,k} - \tau_{0,k-1}) \tag{10}$$

Where:

$$\tau_{0,k} = \int_0^{r_{0,k}} p(r)dr \approx \frac{1}{N}\int_0^{r_{0,k}} \sum_{n=1}^N K(\|r - r_n\|)dr =$$

$$= \frac{1}{N}\sum_{n=1}^N \left( ndtr\left(\frac{r_{0,k} - r_n}{\sqrt{cov}}\right) - ndtr\left(\frac{-r_n}{\sqrt{cov}}\right)\right) \tag{11}$$

Where $ndtr$ expresses the Gaussian cumulative distribution function and the last row in Equation 10 exploits the trapezoidal rule for integrals computation.

## A.2 Derivations of the gradients of AUCOC

$$\frac{d}{d\theta}\mathcal{A} = \int_0^1 \frac{d}{d\theta}\left\{\int_{r_0}^1 \mathbb{E}[c|r]p(r)dr\right\}\frac{d\tau_0}{1 - \tau_0}$$

Here, we use the assumption discussed in Section Methods that $\tau_0$ does not depend on any parameter, thus allowing us to apply Leibnitz's integration rule, obtaining:

$$\frac{d}{d\theta}\mathcal{A} = \int_0^1 \frac{d}{d\theta}\left\{\int_{r_0}^1 \mathbb{E}[c|r]p(r)dr\right\}\frac{d\tau_0}{1 - \tau_0} \tag{12}$$

$$= \int_0^1 \left\{\int_{r_0}^1 \frac{d}{d\theta}\mathbb{E}[c|r]p(r)dr - \mathbb{E}[c|r_0]p(r_0)\frac{dr_0}{d\theta}\right\}\frac{d\tau_0}{1 - \tau_0} \tag{13}$$

$\tau_0$ can be expressed as:

$$\tau_0 = p(r \leq r_0) = \int_0^{r_0} p(r)dr \tag{14}$$

Consequently:

$$\frac{d\tau_0}{d\theta} = \int_0^{r_0} \frac{dp(r)}{d\theta}dr + p(r_0)\frac{dr_0}{d\theta} = 0$$

$$\frac{dr_0}{d\theta} = -\frac{\int_0^{r_0} \frac{dp(r)}{d\theta}dr}{p(r_0)} \tag{15}$$

Plugging this expression back into Equation 12 we obtain:

$$\frac{d\mathcal{A}}{d\theta} = \int_0^1 \left\{\int_{r_0}^1 \frac{d}{d\theta}\mathbb{E}[c|r]p(r)dr + \mathbb{E}[c|r_0]\int_0^{r_0}\frac{dp(r)}{d\theta}dr\right\}\frac{d\tau_0}{1 - \tau_0} \tag{16}$$

Assuming the use of a Gaussian kernel:

$$K(\|r - r_n\|) = \frac{1}{\sqrt{2\pi\alpha}}\cdot\exp\left(-\frac{(r - r_n)^2}{2\alpha^2}\right) \tag{17}$$

And re-writing:

$$\frac{\mathbb{E}[c|r_0]p(r_0)}{p(r_0)} \approx \frac{\frac{1}{N}\sum_{n=1}^N \mathbf{1}(c_n)K(\|r_0 - r_n\|)}{\frac{1}{N}\sum_{n=1}^N K(\|r_0 - r_n\|)} \tag{18}$$

The gradient of the area becomes:

$$
\begin{aligned}
\frac{d\mathcal{A}}{dr_n} &= \int_0^1 \left\{ \int_{r_0}^1 \frac{d}{dr_n} \mathbb{E}[c|r] p(r) dr + \mathbb{E}[c|r_0] \int_0^{r_0} \frac{dp(r)}{dr_n} dr \right\} \frac{d\tau_0}{1-\tau_0} = \\
&= \int_0^1 \Big\{ \int_{r_0}^1 \frac{d}{dr_n} \frac{1}{\sqrt{2\pi}\alpha} \frac{1}{N} \sum_{n=1}^N \mathbf{1}(c_n) \cdot \exp\left(-\frac{(r-r_n)^2}{2\alpha^2}\right) dr + \\
&\quad \mathbb{E}[c|r_0] \int_0^{r_0} \frac{d}{dr_n} \frac{1}{\sqrt{2\pi}\alpha} \frac{1}{N} \sum_{n=1}^N \exp\left(-\frac{(r-r_n)^2}{2\alpha^2}\right) dr \Big\} \frac{d\tau_0}{1-\tau_0} = \\
&= \int_0^1 \Big\{ \int_{r_0}^1 \frac{1}{\sqrt{2\pi}\alpha^3 N} \mathbf{1}(c_n) \cdot (r-r_n) \cdot \exp\left(-\frac{(r-r_n)^2}{2\alpha^2}\right) dr + \\
&\quad \mathbb{E}[c|r_0] \int_0^{r_0} \frac{1}{\sqrt{2\pi}\alpha^3 N} (r-r_n) \cdot \exp\left(-\frac{(r-r_n)^2}{2\alpha^2}\right) dr \Big\} \frac{d\tau_0}{1-\tau_0} = \\
&= \int_0^1 \Big\{ -\frac{1}{\sqrt{2\pi}\alpha N} \mathbf{1}(c_n) \cdot \left[ \exp\left(-\frac{(1-r_n)^2}{2\alpha^2}\right) - \exp\left(-\frac{(r_0-r_n)^2}{2\alpha^2}\right) \right] - \\
&\quad \mathbb{E}[c|r_0] \frac{1}{\sqrt{2\pi}\alpha N} \left[ \exp\left(-\frac{(r_0-r_n)^2}{2\alpha^2}\right) - \exp\left(-\frac{(-r_n)^2}{2\alpha^2}\right) \right] dr \Big\} \frac{d\tau_0}{1-\tau_0}
\end{aligned}
\tag{19}
$$

Also for the gradients in the code implementation we exploited the trapezoidal rule for the computation of the external integral between [0,1].

## B   Potential upper bound to AUCOCLoss

AUCOC has a formulation of the kind $-log(\sum_n z_n)$ with $z_n \in [0,1]$. In contrast, cross-entropy is a $-\sum_n \log z_n$, where contribution of increasing low $z_n$'s to the loss is much larger, hence gradient-based optimization is faster. Exploiting Jensen's inequality, one could find the following upper bound of AUCOCLoss to minimise. However, from thorough experiments it has been proven not to be a tight enough bound for the optimisation to be successful. In fact, trying to optimise the rightmost term of the following equation, instead of the correct definition on the left, does not lead to a satisfactory optimisation of AUCOC.

$$
\begin{aligned}
-\log &\left( \int_0^1 \left\{ \int_{r_0}^1 r_n^* K(||r-r_n||) dr \right\} \frac{d\tau_0}{1-\tau_0} \right) \leq \\
&- \int_0^1 \left\{ \int_{r_0}^1 \log(r_n^*) K(||r-r_n||) dr \right\} \frac{d\tau_0}{1-\tau_0}
\end{aligned}
$$

## C   Training details

In CIFAR100 experiments, we followed Karandikar et al. [16] and used Wide-Resnet-28-10 [41] as the network architecture. We trained the models for 200 epochs, using Stochastic Gradient Descent (SGD), with batch of 512, momentum of 0.9 and an initial learning rate of 0.1, decreased after 60, 120, 160 epochs by a factor of 0.1. We set these parameters based on the best validation performance of CE and we keep it for all the losses. In Tiny-ImageNet experiments, we used ResNet-50 [10] as backbone architecture, SGD as optimiser with a batch size of 512, momentum of 0.9 and base learning rate of 0.1, divided by 0.1 at 40th and 60th epochs as in Mukhoti et al. [26] In DermaMNIST, RetinaMNIST and TissueMNIST [38] experiments, we followed the training procedures of the original paper, employing a ResNet-50 He et al. [10], Adam optimizer. The batch size is set to 128 for DermaMNIST and RetinaMNIST and to 512 for the larger TissueMNIST. We used the initial learning rate 0.0001 for DermaMNIST and 0.001 for RetinaMNIST and TissueMNIST, and trained the models for 100 epochs by reducing the learning rate by 0.1 after epochs 50 and 75.

All the models have been trained using either the NVIDIA GeForce RTX 2080 Ti or NVIDIA GeForce RTX 3090. The datasets have been split as follows.

For CIFAR100 we used 45000/5000/10000 images respectively as training/validation/test sets.

Tiny-ImageNet is a subset of ImageNet with $64 \times 64$ images and 200 classes. We employed 90000/10000/10000 images as training/validation/test set, respectively.

DermaMNIST is composed of dermatoscopic images with 7007/1003/2005 samples for training, validation and test set, respectively, categorised in 7 different diseases.

RetinaMNIST is based on the DeepDRiD24 challenge, which provides a dataset of 1,600 retina fundus images. The task is ordinal regression for 5-level grading of diabetic retinopathy severity. Consistently with the original paper, we split the source training set with a ratio of $9:1$ into training and validation set, and use the source validation set as the test set.

# D    Results on a third medical dataset, TissueMNIST

Tables 5 and 6 report results on an additional medical dataset, TissueMNIST. The dataset contains 236,386 human kidney cortex cells, segmented from 3 reference tissue specimens and organized into 8 categories. We split the source dataset with a ratio of $7:1:2$ into training, validation and test set. The results are consistent with those reported in the main paper.

Table 5: Test results on TissueMNIST for AUCOC and accuracy. The last two columns report $\tau_0$ corresponding to 90% and 95% accuracy. In bold and underlined respectively the best and second best results for each metric.

| Loss funct. | AUCOC ↑ | Acc. ↑ | $\tau_0 \downarrow$ @ acc. 90% | 95% |
|---|---|---|---|---|
| CE | 65,91 | 83,34 | 68,60 | 81,95 |
| FL ($\gamma$=3) | 66,07 | 82,77 | 79,41 | 87,82 |
| AdaFL53 | 66,15 | 83,15 | 75,01 | 89,90 |
| CE+MMCE | 66,52 | 84,12 | 68,81 | 86,84 |
| FL+MMCE | 66,30 | 82,85 | 68,96 | 81,70 |
| CE+S-AvUC | 62,42 | 80,88 | 73,00 | 86,03 |
| FL+ S-AvUC | 62,75 | 79,75 | 74,79 | 84,74 |
| CE+S-ECE | 65,58 | 83,53 | 70,67 | 83,84 |
| FL+S-ECE | 65,20 | 82,59 | 70,90 | 84,06 |
| **CE+AUCOCL** | **67,16** | **84,54** | 68,60 | **77,80** |
| **FL+AUCOCL** | 67,04 | 83,46 | **67,21** | 81,55 |

Table 6: Test results on TissueMNIST for calibration: expected calibration error (ECE), KS score, Brier score and class-wise ECE (cwECE), post TS. In bold and underlined respectively the best and second best results for each metric. Noticeably, AUCOCL performs comparably to the baselines, even though it does not aim at improving calibration explicitly.

| Loss funct. | ECE↓ | KS↓ | Brier↓ | cwECE↓ |
|---|---|---|---|---|
| CE | **0,89** | **0,36** | 46,11 | 1,65 |
| FL ($\gamma$=3) | 2,06 | 0,67 | 46,50 | 1,58 |
| AdaFL53 | 2,10 | 1,11 | 46,58 | 1,79 |
| CE+MMCE | 0,98 | 0,53 | 45,27 | 1,83 |
| FL+MMCE | 1,55 | 0,57 | 45,87 | **1,21** |
| CE+S-AvUC | 2,28 | 1,87 | 50,32 | 1,62 |
| FL+ S-AvUC | 2,49 | 1,83 | 50,99 | 2,00 |
| CE+S-ECE | 2,38 | 0,98 | 52,88 | 2,71 |
| FL+S-ECE | 2,45 | 1,40 | 47,29 | 1,83 |
| **CE+AUCOCL** | 1,07 | 0,71 | **44,41** | 1,29 |
| **FL+AUCOCL** | 3,56 | 0,68 | 45,54 | 1,47 |

# E    Class imbalance experiments

In Table 7 we report the results for accuracy and AUCOC on the widely employed Long-Tailed CIFAR100. We employed a tunable data imbalance ratio (Nmax / Nmin, where N is number of samples in each class), which controls the class distribution in the training set and trained the models with three levels of imbalance ratio, namely 100, 50, 10. Noticeably, when AUCOCLoss complements cross-entropy it obtains the best results for both metrics in all the three settings, with higher benefits at increasing imbalance, while with FL it obtains always the second best AUCOC and comparable accuracy.

Table 7: Accuracy and AUCOC results on CIFAR100 Long-Tailed for 3 degrees of class imbalance in the training set, namely 100, 50 and 10. In bold and underlined respectively the best and second best results for each metric.

| Imbalance ratio | 100 | | 50 | | 10 | |
|---|---|---|---|---|---|---|
| Loss funct. | AUCOC ↑ | Acc. ↑ | AUCOC ↑ | Acc. ↑ | AUCOC ↑ | Acc. ↑ |
| CE | 72,33 | 47,27 | 78,02 | 53,83 | 89,59 | 71,15 |
| FL ($\gamma$=3) | 73,66 | 47,00 | 79,53 | 53,50 | 91,91 | 71,59 |
| AdaFL53 | 73,72 | 47,07 | 79,55 | 53,55 | 92,05 | 71,61 |
| CE+MMCE | 72,92 | 47,80 | 78,54 | 53,45 | 90,66 | 71,32 |
| FL+MMCE | 73,49 | 46,78 | 79,50 | 53,32 | 91,70 | 71,71 |
| CE+S-AvUC | 73,08 | 46,62 | 76,15 | 54,45 | 91,72 | 71,60 |
| FL+ S-AvUC | 72,68 | 46,62 | 78,51 | 53,11 | 89,75 | 72,11 |
| CE+S-ECE | 73,53 | 47,58 | 79,31 | 53,84 | 91,63 | 71,67 |
| FL+S-ECE | 72,30 | 46,13 | 78,56 | 52,84 | 91,29 | 70,6 |
| **CE+AUCOCL** | **75,85** | **49,63** | **81,38** | **55,89** | **92,59** | **72,40** |
| **FL+AUCOCL** | 74,51 | 46,91 | 79,72 | 53,64 | 92,21 | 71,92 |

# F    Calibration results for Tiny-ImageNet and CIFAR100

In Table 8 we report calibration results respectively for CIFAR100 and Tiny-ImageNet. For all the metrics and datasets, AUCOCLoss provides comparable results with respect to the baselines.

Table 8: Test results on CIFAR100 and Tiny-Imagenet for calibration metrics, namely expected calibration error (ECE), KS score, Brier score and class-wise ECE (cwECE), post temperature scaling. In bold and underlined respectively the best and second best results for each metric. Noticeably, AUCOCL performs comparably to the baselines, even though it does not aim at improving calibration explicitly.

| Dataset | CIFAR100 | | | | Tiny-Imagenet | | | |
|---|---|---|---|---|---|---|---|---|
| Loss funct. | ECE↓ | KS↓ | Brier↓ | cwECE↓ | ECE↓ | KS↓ | Brier↓ | cwECE↓ |
| CE | 2,41 | 1,1 | 33,91 | 0,211 | 1,54 | **0,74** | 66,25 | 0,163 |
| FL ($\gamma$=3) | 1,92 | 1,46 | 31,16 | 0,184 | 1,46 | 1,14 | 65,91 | 0,152 |
| AdaFL53 | 1,47 | 1,22 | 31,3 | 0,183 | **1,35** | 0,85 | 65,71 | 0,159 |
| CE+MMCE | 2,32 | 1,23 | 34,6 | 0,209 | 1,79 | 0,93 | 66,42 | 0,157 |
| FL+MMCE | 1,92 | 1,82 | 31,24 | 0,181 | 1,94 | 1,63 | 65,49 | 0,151 |
| CE+S-AvUC | 3,23 | 0,53 | 31,71 | 0,199 | 2,13 | 1,15 | 65,20 | **0,146** |
| FL+ S-AvUC | 1,83 | 0,53 | 31,74 | 0,177 | 1,51 | 1,13 | 64,62 | 0,15 |
| CE+S-ECE | 3,85 | 1,3 | 31,85 | 0,197 | 1,66 | 0,81 | 66,14 | 0,15 |
| FL+S-ECE | 1,74 | 1,56 | 32,19 | 0,180 | 2,6 | 2,37 | 66,40 | 0,161 |
| **CE+AUCOCL** | 1,65 | **0,75** | **29,78** | 0,184 | 1,65 | 1,29 | **64,23** | 0,157 |
| **FL+AUCOCL** | **1,34** | 0,95 | 30,24 | **0,175** | 1,86 | 1,29 | 64,45 | 0,155 |

## G  Results for varying batch size

In Table 9 we report the results on DermaMNIST with reduced batch sizeon which KDE is applied on, i.e. 64 and 28 samples per batch. The results of AUCOCLoss are not significantly affected by it, when complementing cross-entropy and focal loss.

Table 9: Test results on DermaMNIST for accuracy, AUCOC and ECE for batch sizes 64 and 32. Reducing the batch size does not affect significantly the performance of AUCOCL.

| Batch | Loss funct. | AUCOC ↑ | Acc. ↑ | ECE ↓ |
|---|---|---|---|---|
| 64 | **CE+AUCOCL** | 91,16 | 75,18 | 11,00 |
|    | **FL+AUCOCL** | 91,13 | 74,71 | 9,35 |
| 32 | **CE+AUCOCL** | 90,14 | 73,80 | 15,50 |
|    | **FL+AUCOCL** | 91,10 | 73,47 | 6,69 |

## H  Additional results for CIFAR100 and Tiny-ImageNet

In Table 10 we report test results on CIFAR100 using ResNet-50 and Tiny-Imagenet using WideResnet-28-10, run for one seed. Best and second best results are in bold and underlined. They show consistency with the main experiments reported in the paper.

Table 10: Test results on CIFAR100 using ResNet-50 and Tiny-Imagenet using WideResnet-28-10, run for one seed. Best and second best results are in bold and underlined.

| Dataset | CIFAR100 | | | | Tiny-ImageNet | | | |
|---|---|---|---|---|---|---|---|---|
| | | | $\tau_0 \downarrow$ @ acc. | | | | $\tau_0 \downarrow$ @ acc. | |
| Loss funct. | AUCOC ↑ | Acc. ↑ | 90% | 95% | AUCOC ↑ | Acc. ↑ | 65% | 75% |
| CE | 91,10 | 73,69 | 34,72 | 49,42 | 74,03 | 49,29 | 34,74 | 52,28 |
| FL ($\gamma$=3) | 92,68 | 75,13 | 32,30 | 46,40 | 75,05 | 49,56 | 34,17 | 51,72 |
| AdaFL53 | 92,65 | 74,5 | 32,01 | 45,78 | 75,36 | 49,49 | 33,77 | 50,76 |
| CE+MMCE | 92,55 | 74,75 | 31,08 | 44,53 | 74,12 | 48,56 | 35,83 | 52,48 |
| FL+MMCE | 92,46 | 74,47 | 31,69 | 46,63 | 74,79 | 49,32 | 34,57 | 52,12 |
| CE+S-AvUC | 92,63 | 74,12 | 31,02 | 44,83 | 74,14 | 49,13 | 35,00 | 53,17 |
| FL+ S-AvUC | 92,71 | 74,82 | 31,11 | 44,51 | 74,21 | 49,02 | 35,22 | 53,12 |
| CE+S-ECE | 92,57 | 74,27 | 31,38 | 45,4 | 75,62 | 49,91 | 31,88 | 49,55 |
| FL+S-ECE | 92,63 | 75,00 | 31,35 | 47,42 | 74,57 | 49,16 | 35,60 | 52,36 |
| **CE+AUCOCL** | **93,81** | **75,94** | **28,39** | **42,61** | **76,64** | **51,40** | **29,97** | **47,65** |
| **FL+AUCOCL** | 93,19 | 75,90 | 29,45 | 43,11 | 75,86 | 50,84 | 32,00 | 50,12 |

## I  Further explanation on how improve AUCOC

There are two factors which contribute to an increase in AUCOC: decrease in the number of samples delegated to human experts (given the same network accuracy) and increase in the accuracy for the samples that are not delegated but analysed only by the network (given the same human workload).

These two aspects could manifest either individually, if the AUCOC improvement is generated by just a shift "up" or "left" of COC, or in a combined way. The example provided in Figure 1a shows an improvement in both axes ("and" case) and the proposed loss function does not favour one specific behaviour. Figure 2 provides an example of shifts "up" and "left" ("or" cases). From the AUCOC metrics alone, it is not possible to infer which mechanism is taking place.

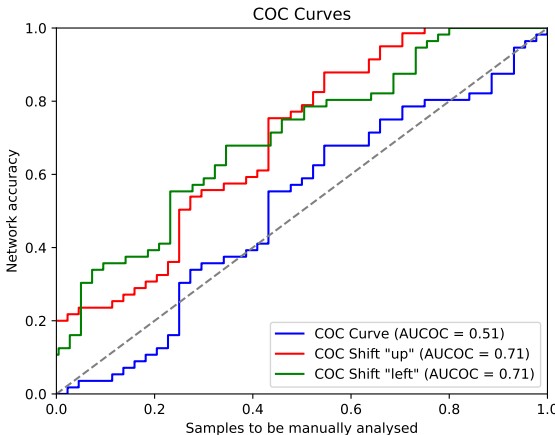

Figure 2: Toy example of possible AUCOC increase by shifting "up" or "left" the blue COC curve.

## J  OOD results on all the perturbations on CIFAR100-C

In table 11 we report the average results on CIFAR100-C for all the perturbations available from Hendrycks and Dietterich [12].

Table 11: Test AUROC(%) on OOD detection, training on CIFAR100 and testing on CIFAR100-C (all 15 perturbations) for MSP, ODIN, MaxLogit and EBM. Best and second best results are in bold and underlined.

| Dataset Loss funct. | **C100-c all corruptions** AUROC↑ | | | |
| --- | --- | --- | --- | --- |
| | MSP | ODIN | MaxLogit | EBM |
| CE 66,18 | 66,42 | 67,71 | 67,71 | |
| FL ($\gamma$=3) | 66,57 | 66,36 | 66,69 | 66,42 |
| AdaFL53 | 66,58 | 66,33 | 66,94 | 66,73 |
| CE+MMCE | 66,12 | 66,22 | 66,34 | 66,23 |
| FL+MMCE | 66,92 | 66,14 | 66,97 | 66,60 |
| CE+S-AvUC | 67,29 | 67,45 | 67,93 | 67,96 |
| FL+ S-AvUC | 66,78 | 66,00 | 67,38 | 67,24 |
| CE+S-ECE | 67,32 | 67,50 | 67,98 | 68,01 |
| FL+S-ECE | 67,34 | 65,38 | 67,38 | 67,78 |
| **CE+AUCOCL** | 67,77 | **67,77** | 69,51 | **69,69** |
| **FL+AUCOCL** | **68,00** | 66,83 | **69,52** | 69,46 |

