# OpenReview forum: "Expert load matters: operating networks at high accuracy and low manual effort"
_NeurIPS.cc/2023/Conference — NeurIPS 2023 poster_

### Official Review · Reviewer_jAPT · 2023-07-01

**Soundness:** 4 excellent
**Presentation:** 4 excellent
**Contribution:** 3 good
**Rating:** 7
**Confidence:** 4

**Summary:**

This work supposes a real-world setting where misclassified examples are reviewed post-hoc by human experts, and offers a near-optimal trade-off between such examples (the expert load) and classifier accuracy. The authors propose to use a curve of confidence versus expert load, using the latter as a sliding scale to arrive at that optimum. They propose a no-binning method that kernelises the said (CFOC) curve, and arrive at a formulation to minimize the expert load value irrespective of the network parameters so that the formulation may be added to the error minimisation loop.

Extensive experiments are performed, such as those on OOD detection where overconfidence is significant, and imbalance, where traditional OC curves are less meaningful because the underlying metrics are too.


**Strengths:**

The idea has a high likelihood of orginality and novelty, to the extent that benchmarking has had to be done with the COC-derived-loss-augmented CE and all sorts of CE-derived losses, and not competing methods. The idea is developed in a simple manner that persuades the reader's logic. Experimentation is fairly complete, even though a discussion of mixture ratios of CE and AUCOC- losses was expected too.


**Weaknesses:**

Studying the dynamic of mixing two losses that have such variable gradient magnitude properties on the rate and preciseness of convergence is something I'd have done.

The $\tau$ being the bandwidth of the kernel, seems to be extraordinarily large at both levels in table 1. I'm not sure if I'd be offloading up to a third of testing examples to the expert.


**Questions:**

Equation 3 does not mention how AUCOC is determined. It just computes the AUCOC Loss using the former. Please let me know in my assuming it is determined in the same way as the ECE does, i.e.  accuracy vs. confidence, as is strongly evident from the presence of bins? How do you define confidence?


**Limitations:**

yes

---

> ### Author Rebuttal · Authors · 2023-08-09
>
> We thank the reviewer for appreciating the proposed method and we are happy to answer to the raised questions.
>
> AUCOCLoss (Equation 3), exploits the definition of AUCOC (equation in Section 3.2.2). Specifically, to build a differentiable loss out of this definition, we employ KDE to define E[c|r]p(r) and tau0 as explained in Section 3.3.1.
> The metrics AUCOC instead, as it does not need to be differentiable, is just built in the same fashion as classic AUROC, i.e. as explained in lines 215-224.
>
> Regarding the study about mixing the two losses, it is definitely an interesting point. We tried various weighting factors and found out that favourable weighting factors for AUCOCLoss fell all between 1 and 10, without no unified preferred value in this range, but with consistent improvement over the baselines.

---

> > ### Comment · Area_Chair_f3E5 · 2023-08-19
> >
> > Thank you for the response, we will take the additional explanations into consideration for further discussions,
> >
> > Best regards,
> > AC

---

### Official Review · Reviewer_eeZS · 2023-07-03

**Soundness:** 2 fair
**Presentation:** 2 fair
**Contribution:** 2 fair
**Rating:** 5
**Confidence:** 3

**Summary:**

The authors present a "confidence operating characteristic" curve to represent tradeoff between accuracy and numbers of samples delegated to human experts. To maximize the area under this curve, the authors propose a new loss. The authors run classification experiments on computer vision and medical image datasets.

**Strengths:**

The paper tackles a problem that is clearly important -- namely, minimizing the human effort needed in a human-in-the-loop system and quantifying the "human effort" portion. However, see weaknesses section below.

**Weaknesses:**

- The references list looks reasonable (35 references), but for whatever reason, the related works section seems very sparse and lacking in relevant bodies of work. Upon closer inspection, it looks like a significant portion of those references (10+) are not related, just part of the motivation. This work is effectively "active learning," which is conspicuously absent from the paper entirely. I'm not super familiar with active learning, but no doubt there is previous work that already tackles the problems introduced here -- and it's not clear how this paper is different. For example, here's a highly cited paper in the area: "Cost-Effective Active Learning for Deep Image Classification" 2017, https://arxiv.org/abs/1701.03551. The paper here proposes a metric that minimizes annotation cost -- in particular, they ask humans to annotate highly-unconfident samples. This seems like a reasonable metric and it's not clear why end-to-end minimizing cost is better, for example. (I'm sure it could be justified) Granted, the setup is slightly different, but I'm sure a more thorough search will yield more relevant papers.
- The paper doesn't cite previous metrics for "human effort", how they are fallible, and how the proposed metric fixes that problem. As a result, it's not clear why this metric is preferred to other variants.
- Experiments are performed on MNIST variants, CIFAR100 and TinyImageNet, but do these results extend to ImageNet for example? A full ImageNet run isn't needed; just a few epochs showing that results are trending positively for your method.

**Questions:**

See weaknesses above.

**Limitations:**

See weaknesses above.

---

> ### Author Rebuttal · Authors · 2023-08-09
>
> # Comment about active learning and related work.
>
> We believe there is a misunderstanding, as the proposed paper is not performing active learning. Active learning and our work have two inherently different goals. Active learning aims to include the human expert in the loop **during training**. In contrast, our work focuses on reducing the load on human expert **at deployment**, i.e., when the model is used for inference. In many applications, e.g., medical imaging, to meet high accuracy requirements, samples for which a model is not certain are delegated to experts. Our goal is to find training losses that will take into account this delegation.
>
> Consequently, our study has primarily concentrated on a different literature than active learning, namely confidence calibration, that also explicitly focuses on predictive confidences and ultimately aims to enhance human-AI interaction. We outlined the most recent and relevant methodologies in this realm. Moreover, our investigation explores the allocation of samples for human analysis during deployment. This quantity is a recognised metric, and analogous curves resembling COC have been previously employed, as in [1]. In response to the reviewer's suggestion, to further improve our work, we will incorporate supplementary references to exemplar applications that utilise akin metrics [2, 3, 4].
>
> To avoid any future misunderstandings or doubts, we will also add a discussion to the main paper where we explicitly state the differences with active learning.
>
> # Comment about "human effort" metrics.
>
> To the best of our understanding of this comment, we believe this is related to the misunderstanding about the relation between this work to active learning. In this work we are only interested in the number of samples delegated to a human expert for analysis **at deployment time**. This number is already an established metric and curves similar to COC have been used before [1]. Following the reviewer's suggestion, we can add additional citations to example applications using similar quantities [2,3,4]. However, to the best of our knowledge, this is the first work that incorporates AUCOC in a loss function in a differentiable way. We acknowledge that in active learning, "human effort" may be quantified in different ways, but this work is fundamentally different than active learning, as we discuss in detail as response to the previous concern of the reviewer.
>
> [1] Gorski, N., V. Anisimov, E. Augustin, O. Baret, and S. Maximov (2001). Industrial bank check processing: the a2ia checkreadertm. International Journal on Document Analysis and Recognition 3(4), 196–206.\
> [2] Dvijotham, K.(., Winkens, J., Barsbey, M. et al. Enhancing the reliability and accuracy of AI-enabled diagnosis via complementarity-driven deferral to clinicians. Nat Med 29, 1814–1820 (2023). \
> [3] Leibig, C. et al. Combining the strengths of radiologists and AI for breast cancer screening: a retrospective analysis. Lancet Digit. Health 4, e507–e519 (2022).\
> [4] Hendrycks, D. & Gimpel, K. A baseline for detecting misclassified and out-of-distribution examples in neural networks. In Proceedings of International Conference on Learning Representations (ICLR) (OpenReview.net, 2017).
>
> # ImageNet preliminary results.
>
> Larger-scale datasets are indeed very interesting. However, resource and energy consumption of experimenting with such datasets at the breadth we present - which requires training from scratch and assessing multiple aspects of the model with different data sets - is extremely high. Following the reviewer's suggestion, we report preliminary results on ImageNet in Table 3 of the PDF uploaded in the "global" rebuttal response, running AUCOCLoss and the cross-entropy baseline for 15 epochs with ResNet-50. Noticeably, compared to the baseline, AUCOCLoss shows favourable preliminary results.

---

> > ### Comment · Reviewer_eeZS · 2023-08-14
> > **Thanks for the clarifications**
> >
> > Thanks to the authors for a thorough explanation:
> >
> > - The distinction between train-time and deployment-time makes a lot of sense. I now see why the related works section mentions the fields that it does.
> > - It's also helpful to know these metrics are already in-use and established; I missed that in the paper before.
> > - I also appreciate the ImageNet run. The preliminary results look convincing to me.
> >
> > I've bumped by rating from Reject to Borderline accept, as my old rating was based on a misunderstanding of the paper's contributions.

---

### Official Review · Reviewer_1tYR · 2023-07-07

**Soundness:** 2 fair
**Presentation:** 3 good
**Contribution:** 2 fair
**Rating:** 4
**Confidence:** 4

**Summary:**

The paper aims to address the trade-off between model accuracy and model confidence results. The authors propose a novel loss function called AUCOC, which maximizes the area under the confidence operating characteristic curve. They evaluate the performance of their approach on various image classification and medical image classification datasets.

**Strengths:**

The problem being addressed is of significant importance and has been widely studied.



**Weaknesses:**

1. In terms of empirical evaluation, the authors only assess the performance on image classification datasets. As this is primarily an empirical paper, it would be beneficial to see results on different types of datasets, including text data, tabular data, and other diverse datasets. Moreover, I am curious to know how the proposed model performs on larger-scale datasets, as the ones evaluated in the paper (e.g., CIFAR100 and Tiny-ImageNet) may not be sufficiently large-scale.

2. Furthermore, when it comes to out-of-distribution (OOD) detection results, it would be valuable for the authors to compare their proposed methods with commonly used approaches such as MSP, MaxLogit, and others.

The proposed method is commendably simple and straightforward. However, I believe that a more comprehensive evaluation is necessary to demonstrate the effectiveness of the proposed approach.

**Questions:**

See cons above

**Limitations:**

Yes

---

> ### Author Rebuttal · Authors · 2023-08-09
>
> We thank the reviewer for acknowledging the importance of the task and appreciating the simplicity of the proposed method. We are happy to address the raised concerns.
>
> # Comment about additional datasets.
>
> We would like to point out that the presented article is not primarily an empirical study, as the reviewer suggests. Considerable modelling effort is required to formulate the loss, which is a novel contribution that is appreciated by R1 and R4. We aimed for a clear and easy explanation, and, to this end, we placed all lengthy derivations in the Appendix.
>
> While we appreciate that demonstrations on diverse types of data can be beneficial, there are two main reasons why we refrain from this: (i) the aim here is to present a novel loss and evaluated it consistently with experiments from the baselines [1, 2] and (ii) imaging data is one of the most challenging data types used in many different ML articles. Therefore, we chose to focus on explaining the loss well and demonstrate its different aspects on well established data sets.
>
> Larger-scale datasets are indeed very interesting, however, resource and energy consumption of experimenting with such datasets at the breadth we present - which requires training from scratch and assessing multiple aspects of the model with different datasets - is extremely high. We report some initial results on ImageNet here to address the reviewer's concerns. Results are summarised in Table 3 of the PDF uploaded in the "global" rebuttal response, running AUCOCLoss and the cross-entropy baseline for 15 epochs with ResNet-50. Noticeably, compared to the baseline, AUCOCLoss shows favourable preliminary results in all the metrics.
>
>
>
> [1] Karandikar, A., N. Cain, D. Tran, B. Lakshminarayanan, J. Shlens, M. C. Mozer, and B. Roelofs (2021). Soft calibration objectives for neural networks. In NeurIPS.\
> [2] Mukhoti, J., V. Kulharia, A. Sanyal, S. Golodetz, P. Torr, and P. Dokania (2020). Calibrating deep neural networks using focal loss. In H. Larochelle, M. Ranzato, R. Hadsell, M. Balcan, and H. Lin (Eds.), Advances in Neural Information Processing Systems, Volume 33, pp. 15288–15299. Curran Associates, Inc.
>
> # Comment about additional OOD experiments.
>
> We would like to highlight that the detectors provided in the main paper are already the commonly used approaches: AUROC MSP [3] (pre temperature scaling) and ODIN (post temperature scaling) [4], even though we did not mention the original names explicitly, but only described in lines 275-277. This was also adopted by the baseline [5]. We thank the reviewer for pointing this out, and we will add the names to the paper to make it clearer and more explicit.
>
> Following the reviewer's suggestion, in addition we report the results with two other OOD detectors, MaxLogit [6] and EBM [7], for SVHN, CIFAR-C under Gaussian noise and additionally CIFAR-C under all the 15 corruptions from [8]. Results are summarised in Table 2 of the PDF uploaded in the "global" rebuttal response. The new results are consistent with those presented in the paper i.e., for every detector and dataset, AUCOCLoss provides the best OOD detection performance and, in almost all the cases, also the second-best.
>
> [3] Hendrycks, D., & Gimpel, K. (2016). A baseline for detecting misclassified and out-of-distribution examples in neural networks. ICLR.\
> [4] Hsu, Yen-Chang, Yilin Shen, Hongxia Jin and Zsolt Kira. “Generalized ODIN: Detecting Out-of-Distribution Image Without Learning From Out-of-Distribution Data.” 2020 IEEE/CVF Conference on Computer Vision and Pattern Recognition (CVPR) (2020): 10948-10957.\
> [5] Mukhoti, J., V. Kulharia, A. Sanyal, S. Golodetz, P. Torr, and P. Dokania (2020). Calibrating deep neural networks using focal loss. In H. Larochelle, M. Ranzato, R. Hadsell, M. Balcan, and H. Lin (Eds.), Advances in Neural Information Processing Systems, Volume 33, pp. 15288–15299. Curran Associates, Inc.\
> [6] Hendrycks, D., Basart, S., Mazeika, M., Mostajabi, M., Steinhardt, J., & Song, D. (2022). Scaling out-of-distribution detection for real-world settings. ICML.\
> [7] Liu, W., Wang, X., Owens, J., & Li, Y. (2020). Energy-based out-of-distribution detection. NeurIPS.\
> [8] Hendrycks, D. and T. Dietterich (2019). Benchmarking neural network robustness to common corruptions and perturbations. Proceedings of the International Conference on Learning Representations

---

> ### Author Response · Authors · 2023-08-17
> **Approaching end of the author-reviewer discussion phase**
>
> We would like to kindly draw the attention of the Reviewer that the discussion phase is soon reaching its conclusion. We did our best to thoroughly address the raised concerns and we are eager to engage further, should any additional clarification be beneficial.

---

> > ### Comment · Area_Chair_f3E5 · 2023-08-19
> >
> > Thank you for the response, we will take the additional explanations into consideration for further discussions,
> >
> > Best regards,
> > AC

---

### Official Review · Reviewer_Bcd1 · 2023-07-25

**Soundness:** 3 good
**Presentation:** 3 good
**Contribution:** 3 good
**Rating:** 6
**Confidence:** 3

**Summary:**

This paper proposes a new loss, AUCOC loss, to improve the networks accuracy and prediction confidence. The loss aims to reduce the number of errors made by the algorithm and thus also the number of delegated samples to domain experts.  The proposed loss focuses on maximizing the area under the COC curve during training in a differentiable manner. The AUCOC loss is used complementary to the original network’s training loss, resulting in increased classification accuracy, better OOD sample detections, and on par calibration performance.

**Strengths:**

The usage of the ''area under the confidence operating characteristics'' curve as an additional loss (being differentiable) is a novel idea.

The paper is well written, the core method is well explained and the experimental results back the claims of the paper.

The proposed AUCOC loss is outperforming other loss functions.

The possible weakness of only using AUCOC loss is clearly described.


**Weaknesses:**

The title is somewhat misleading.

A clear reference of which CNNs are used for the experiments is needed, especially since different CNNs were used for the experiments.

How does ResNet-50 behave for CIFAR100 using the AUCOC loss and how does Wide-Resnet-28-10 behave on TinyImageNet? Was the latter Wide-Resnet also used for table 4?

Why was only the corruption type Gaussian noise evaluated for the CIFAR 100-C experiments?

The paper wants to minimize the expert load and to be able to predict OOD samples correctly. So far, the paper shows that the accuracy increases and the amount of delegated samples decreases in the OOD setting. While in its current state, the paper has important contributions, it would be interesting to explore if the proposed method also influences the prediction behavior of the CNN. [1] analyses the prediction behavior of CNNs and finds a shape-bias cue conflict: CNNs tend to recognize texture rather than shape, which is in contrast to human vision behavior, which recognizes objects based on their shapes. Therefore, an obvious step would be to investigate how the networks predict objects (i.e., shape or texture), and if the AUCOC can also influence that behavior.

Line 149-153: This or/and is confusing. When does a higher AUCOC indicate lower number of samples delegated to humans *and* also a higher accuracy, and when is it one of these results?

Typos:

References to Eq. 3 are sometimes misleading, while a reference to the Eq. in line 153 would sometimes be more appropriate.

Line 173: “estimate”: choose

Line 178: Leibniz integral rule

[1] Robert Geirhos, Patricia Rubisch, Claudio Michaelis, Matthias Bethge, Felix A. Wichmann, Wieland Brendel. ImageNet-trained CNNs are biased towards texture; increasing shape bias improves accuracy and robustness. ICLR 2019


**Questions:**

See Weaknesses.

It would be interesting to see if the proposed AUCOC loss can influence the CNN prediction behavior to be more in line with humans resulting in a lower need for experts.

**Limitations:**

The paper addressed the limitations of why the AUCOC loss is rather used in conjunction with classical CNN loss functions, like cross-entropy.

---

> ### Author Rebuttal · Authors · 2023-08-09
>
> We thank the reviewer for appreciating the proposed work and helping us improve the explanations. We are happy to address the raised doubts and questions.
>
>
> # Comment about employed architectures and additional experiments on CIFAR100 and Tiny-ImageNet.
>
> We thank the reviewer for pointing this out. Due to space reasons, we provided information on which architectures have been used in Appendix 7. To be consistent with the baseline [1], we used Wide-Resnet-28-10 for CIFAR100 (OOD in Table 4 as well) and Resnet-50 for Tiny-Imagenet. Following the suggestion of the reviewer, we report results on CIFAR100 with Resnet-50 and on Tiny-Imagenet with Wide-Resnet-28-10 in Table 1 of the PDF uploaded in the "global" rebuttal response. Consistently with the results in the paper, AUCOCLoss is better than all the baselines in terms of accuracy and AUCOC, and it provides lower delegated samples.
>
> [1] Karandikar, A., N. Cain, D. Tran, B. Lakshminarayanan, J. Shlens, M. C. Mozer, and B. Roelofs (2021). Soft calibration objectives for neural networks. In NeurIPS.
>
> # Comment about OOD setup.
>
> We evaluated the proposed method on OOD, among the various tasks. Consistently with our baseline [2], we provided an example of stronger dataset shift with SVHN and a weaker one with CIFAR-C under Gaussian noise. Following the reviewer's suggestion, we report the average AUROC(%) on all the 15 corruptions provided in [3] on CIFAR-C. Results are presented in Table 2 of the PDF uploaded in the "general" rebuttal response. Please note that we explicitly referred to the two detectors already employed in the main paper with their commonly used names, i.e., MSP and ODIN. Following the suggestion of R3, we added MaxLogit and EBM to further consolidate the presented findings. The new results are consistent with those presented in the paper, i.e. AUCOCLoss provides the best OOD detection performance for all the experiments and, in almost all the cases, also the second-best.
>
> We would like to highlight that the current method does not explicitly enforce networks to focus on "human-like" learning, e.g., focusing more on shapes rather than textures, as we are neither directly acting on representation learning, nor providing human feedback to the network during training, therefore we do not expect at the moment an improvement towards this end. However, it is definitely an interesting future investigation, to reduce the need of the expert analysis in a human-AI system.
>
>
> [2] Mukhoti, J., V. Kulharia, A. Sanyal, S. Golodetz, P. Torr, and P. Dokania (2020). Calibrating deep neural networks using focal loss. In H. Larochelle, M. Ranzato, R. Hadsell, M. Balcan, and H. Lin (Eds.), Advances in Neural Information Processing Systems, Volume 33, pp. 15288–15299. Curran Associates, Inc.\
> [3] Hendrycks, D. and T. Dietterich (2019). Benchmarking neural network robustness to common corruptions and perturbations. Proceedings of the International Conference on Learning Representations
>
> # Comment about AUCOC improvement explanation (lines 149-153).
>
> We thank the reviewer for drawing attention to it, thus helping clarify the explanation. In lines 149-153 we aim to explain what an improvement, i.e. increase, in AUCOC could practically correspond to. In that paragraph, we wanted to clarify that there are two factors which contribute to an increase in AUCOC: decrease in the number of samples delegated to human experts (given the same network accuracy) and increase in the accuracy for the samples that are not delegated but analysed only by the network (given the same human workload).
>
> These two aspects could manifest either individually, if the AUCOC improvement is generated by just a shift "up" or "left" of COC, or in a combined way. Hence the "or/and". The example provided in Figure 1a of the paper shows an improvement in both axes ("and" case) and the proposed loss function does not favour one specific behaviour. Figure 1 in the PDF uploaded in the "global" rebuttal response provides an example of  shifts "up" and "left" ("or" cases). From the AUCOC metrics alone, it is not possible to infer which mechanism is taking place.

---

> ### Author Response · Authors · 2023-08-17
> **Approaching end of the author-reviewer discussion phase**
>
> We would like to kindly draw the attention of the Reviewer that the discussion phase is soon reaching its conclusion. We tried to thoroughly address the raised concerns and we are eager to engage further, should any additional clarification be beneficial.

---

> > ### Comment · Reviewer_Bcd1 · 2023-08-18
> >
> > I appreciate the author's response and their additional evaluations.
> > Thus, I will increase my score.

---

### Author Rebuttal · Authors · 2023-08-09

We thank the reviewers for taking the time to read our paper and for providing insightful feedback and input.

In each rebuttal we addressed the individual concerns of the reviewers and we are happy to respond to any additional doubts or questions.
In the PDF uploaded in this "general" response, we provide the following additional results and clarifying figures, following reviewers' suggestions:
- Additional experiments on CIFAR100 and Tiny-Imagenet;
- Additional OOD experiments, with new OOD detectors and more  CIFAR-C corruptions;
- Preliminary ImageNet results;
- A toy example of AUCOC improvement.

---

### Decision · Program_Chairs · 2023-09-21

**Decision:**

Accept (poster)

**Comment:**

This paper assumes a setting where misclassified examples are reviewed by human experts for model analysis at deployment time, and attempts to minimize the human expert load. It proposes an AUCOC loss to improve the networks accuracy and prediction confidence, where the AUCOC loss aims to reduce he number of samples to be reviewed by to domain experts. The proposed loss maximizes the area under the COC curve. Used in addition to the respective original training loss, AUCOC can provide increased classification accuracy, OOD sample detections, and on par calibration performance.
While the paper was generally well received by the reviewers, there was some initial criticism for example due to the experimental evaluation. This was however addressed during the rebuttal phase (including the concerns of reviewer 1tYR w.r.t. the comparison to existing methods) so that after the rebuttal and discussion three out of four reviewers are in favor of accepting the paper. The proposed approach is novel and interesting and the paper is well written. The authors are encouraged to include the feedback from the rebuttal phase into the final version of their submission.